# Proton-dynamic therapy following photosensitiser activation by accelerated protons demonstrated through fluorescence and singlet oxygen production

M. Grigalavicius [1], M. Mastrangelopoulou[1], K. Berg[1], D. Arous[1], M. Ménard [1], T. Raabe-Henriksen[1],
E. Brondz[2], S. Siem[2], A. Görgen [2], N.F.J. Edin [2], E. Malinen[2,3] & T.A. Theodossiou [1]

We demonstrate excitation of photosensitisers (PSs) by accelerated protons to produce fluorescence and singlet oxygen. Their fluorescence follows a pattern similar to the proton energy loss in matter, while proton-derived fluorescence spectra match the photon-induced spectra. PSs excited in dry gelatin exhibit enhanced phosphorescence, suggesting an efficient PSs triplet state population. Singlet oxygen measurements, both optically at ~1270 nm and through the photoproduct of protoporphyrin IX (PpIX), demonstrate cytotoxic singlet oxygen generation by proton excitation. The singlet oxygen-specific scavenger 1,4-diazabicyclo [2.2.2]octane (DABCO) abrogates the photoproduct formation under proton excitation, but cannot countermand the overall loss of PpIX fluorescence. Furthermore, in two cell lines, M059K and T98G, we observe differential cell death upon the addition of the PS cercosporin, while in U87 cells we see no effect at any proton irradiation dose. Our results pave the way for a novel treatment combining proton therapy and "proton-dynamic therapy" for more efficient tumour eradication.

[1] Department of Radiation Biology, Institute for Cancer Research, Radium Hospital, Oslo University Hospital, Montebello 0379 Oslo, Norway. [2] Department of Physics, University of Oslo, Blinderrn 0315 Oslo, Norway. [3] Department of Medical Physics, Oslo University Hospital, Montebello 0379 Oslo, Norway. Correspondence and requests for materials should be addressed to T.A.T. (email: ththeo@rr-research.no)

Photodynamic therapy (PDT)[1,2] provides targeted cancer treatment through the synergy of three individually non-chemotoxic components: the photosensitiser (PS), i.e. a light-activated drug; light of the appropriate wavelength to excite the PS; and oxygen as the terminal generator of toxic species upon interaction with the excited PS[3,4]. The photodynamic action is effected through the generation of reactive oxygen species (ROS) either by: (i) charge transfer between PS and molecules, which could involve oxygen superoxide anion and hydrogen peroxide ultimately leading to the formation of hydroxyl radicals[5] (type I mechanism) or (ii) energy transfer, resulting in the generation of deleterious singlet oxygen [$O_2$ ($^1\Delta_g$) or $^1O_2$] (type II mechanism). The deexcitation of singlet oxygen to its triplet ground state, which is a spin forbidden transition, can be detected optically as low intensity phosphorescence at ~1270 nm.

The main limitation of PDT is the penetration depth of light, which can at most reach a few millimetres in tissue[1].

However, proton therapy[6], the treatment of cancer with accelerated protons, can be tumour-specific, as protons with appropriate initial kinetic energy can deposit a high amount of radiant energy at any selected depth. When the proton beam enters and traverses tissue, the protons mainly lose energy by electromagnetic interactions with atomic electrons, initially at an almost constant rate. When the protons reach low energies, they rapidly deposit their remaining energy over a small region called the Bragg peak. This peak is the therapeutic/cytotoxic window that needs to be spatially-matched to the targeted cancerous lesion. The Bragg peak can be tuned to occur over a defined range within the tissue by selecting an appropriate span of initial proton energies. Thus, the depth of penetration of protons can be increased (or decreased) by amplifying (or reducing) the proton energy to include the target lesion in the Bragg peak. The main advantage of protons over high energy therapeutic X-rays is the lack of exit dose, as all their energy is delivered before, and especially at, the Bragg peak, while the main benefit of protons over visible therapeutic light is that protons can excite atoms over a broad energy range because of the continuous electromagnetic interactions with atomic electrons.

In the present study, we show the excitation of organic PSs in solutions or gels by accelerated protons to generate fluorescence and ROS. Most importantly, we present proof of principle of our hypothesis on two glioblastoma multiforme (GBM) cell lines. These findings are the first step towards a novel hybrid cancer treatment combining proton therapy with singlet-oxygen and other ROS-mediated cytotoxicity.

## Results

**Proton-induced fluorescence imaging.** The experimental set-up used for the present work is shown in Fig. 1. The PSs fluorescence induced by protons, was imaged by a digital camera through a red filter. Analysis of the resulting images with FLUKA Monte Carlo software revealed that the spatial fluorescence pattern (Fig. 2) matched the simulated proton dose deposition pattern (Supplementary Fig. 1). The estimated proton energy at the cuvette entrance was 12.5 MeV, giving a simulated Bragg peak at around 1.8 mm in the PS solution. Correspondingly, the maximal fluorescence intensities were found at the depths of 1.6 mm (*meso*-tetrahydroxyphenyl chlorin—mTHPC, 55 μM), 1.6 (Erythrosin B—ErB, 110 μM), and 1.7 mm (protoporphyrin IX—PpIX, 70 μM) from the cuvette entrance interface. In all cases (Fig. 2) there was a deviation between the fluorescence pattern and the proton dose deposition profile (elevated fluorescence prior to the Bragg peak onset), which will be further discussed.

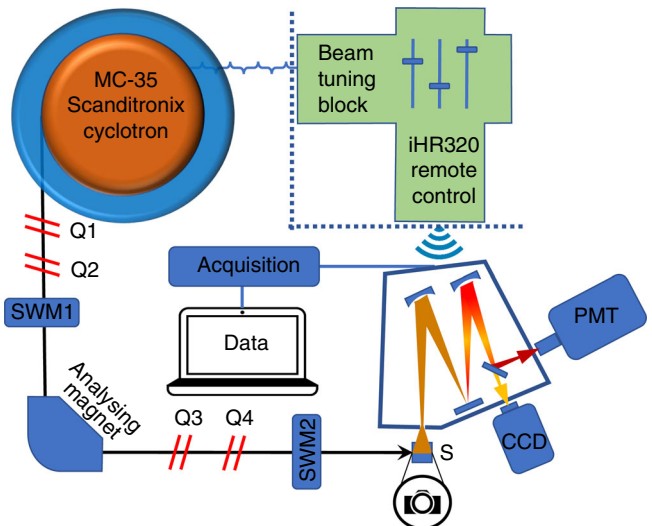

**Fig. 1** Experimental setup. Cyclotron and the main components in the proton beam path to the sample, including quadrupole magnets (Qs) to shape and focus the beam, switching magnets (SWM) to choose the exit port and an analysing magnet to select the energies by bending the beam 90° and the sample (S). The detection system comprised a grating spectrograph coupled to a synapse charge-coupled device (CCD) camera and a near-infrared photomultiplier tube (PMT)

**Proton-induced fluorescence spectra.** Proton-induced fluorescence spectra were also registered in PS solutions and compared to the corresponding spectra obtained by photonic excitation (Fig. 3). The proton-induced emissions in all cases matched spectrally the light-induced fluorescence. From a panel of nine PSs presented in Fig. 3, porphyrin- and chlorin-type PSs gave the strongest fluorescence while quinone-type compounds hyperycin (Hyp) and cercosporin (Cerco) produced weaker fluorescence at the same concentration and proton fluence rate. Of the PSs tested, hypocrellin A generated the weakest signal, observable only at concentrations higher than 20 μM and with 2–4 times increased proton fluence rate (data not shown). We also studied two UV dyes, coumarin 102 and 6,8-difluoro-7-hydroxy-4-methylcoumarin (DiFMU) (Supplementary Fig. 2a, b). As expected, coumarin 102 yielded a ~4-fold higher fluorescence intensity, even though its concentration was 25 times smaller (200 μM vs. 5 mM), since coumarin 102 is a laser dye with a high fluorescence quantum yield in polar solvents[7].

With the same detection system, we studied the fluorescence intensity dependence on PS concentration (Fig. 4a) and proton fluence (Fig. 4b) for selected PSs. Hypericin (Hyp, dimethyl sulfoxide [DMSO]) exhibited a linear response to higher concentrations and was employed to probe the relation between proton-induced fluorescence and proton fluence, which was found to be linear in the range of fluences studied. To determine the most efficient PS for fluorescence generation, the proton-induced fluorescence area was integrated for each PS (10 μM, $8 \times 10^8$ protons $\times$ cm$^{-2}$ $\times$ sec$^{-1}$) and divided by their corresponding integral of the extinction coefficients over the wavelengths from 350 to 750 nm. These values were plotted versus the integrated fluorescence for each PS (Fig. 4c). By normalising the integrated proton-induced fluorescence by the integrated extinction coefficient for each PS, we delineate the dependence on conjugated and resonant π electron systems, as these are responsible for visible light absorbance. In this sense Hyp, Cerco, ErB and Rose Bengal (RoseB) are more efficient in utilising their electrons for interactions with protons, while mTHPC and aluminium 1,4-di

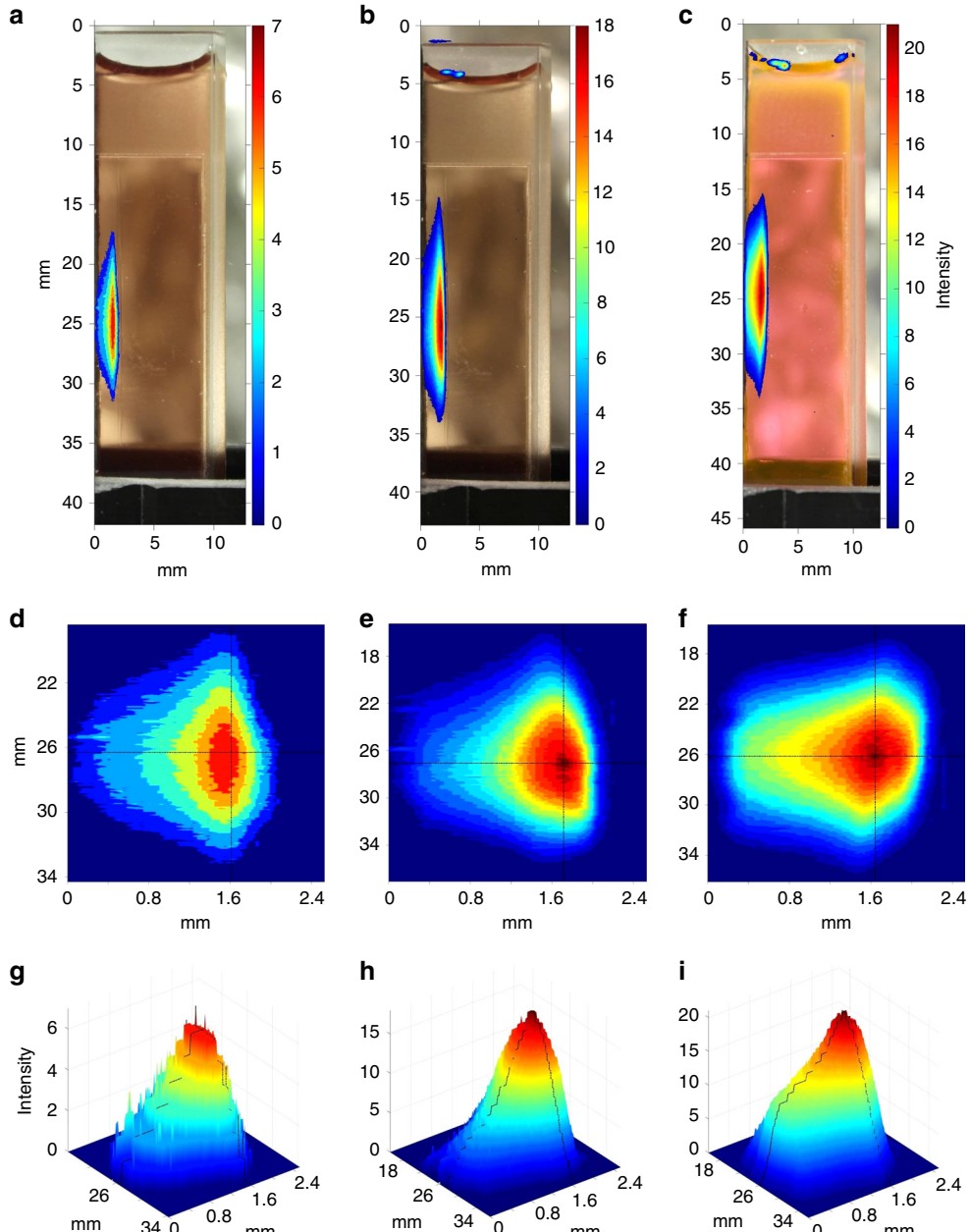

**Fig. 2** Imaging of proton-induced fluorescence. **a–c** Fluorescence images captured during proton irradiation by a Samsung NX1000 camera (F-stop f/5.6, ISO speed 200) after a long pass filter with a cut-off wavelength of 610 nm. The resulting images were processed with MATLAB® to yield heat map representations of the fluorescence emission. **a** mTHPC 55 µM in DMSO. **b** PpIX 70 µM in dimethyl sulfoxide (DMSO) **c** Erythrosin B 110 µM in DMSO. **d–f** Top-view 2D contour heat map representations of **a–c** in an expanded scale. The black lines cross at the estimated position of fluorescence maximum. **g–i** Corresponding 3D representations of (**d–f**). Again, the black lines denote the estimated position of the maximum

(sulfonyloxy)phthalocyanine (AlPcS2a) possess more delocalised electrons and hence can produce more proton-induced fluorescence.

**Proton-induced phosphorescence in dry gels**. Proton-induced luminescence was also studied in anhydrous PS gels made from gelatin, which served as samples depleted of oxygen and with minimal intermolecular collisions (Fig. 5). The absence of collisional quenching (lack of oxygen) in dry gels led to enhanced phosphorescence emissions compared to solutions. The spectral profile and position of the proton-induced fluorescence and phosphorescence emission bands were verified by photonic

excitation. The peak intensity of ErB phosphorescence (Fig. 5b) was found to be at least 40 times higher than for RoseB (Fig. 5a), despite the fact that both sensitisers gave relatively similar intensity phosphorescence under photonic excitation (right axes).

**Proton-induced singlet oxygen generation**. We also registered the production of singlet oxygen from PS solutions by use of the near infrared PMT around ~1270 nm, despite the low intensity of the signal and low signal-to-noise ratio. Singlet oxygen luminescence registration was achieved for ErB and RoseB (Fig. 6a, b, respectively) in acetone d6 (singlet oxygen lifetime in acetone d6 is ~1046 µs vs 3.45 µs in water at 25 °C)[8]. The generation of singlet

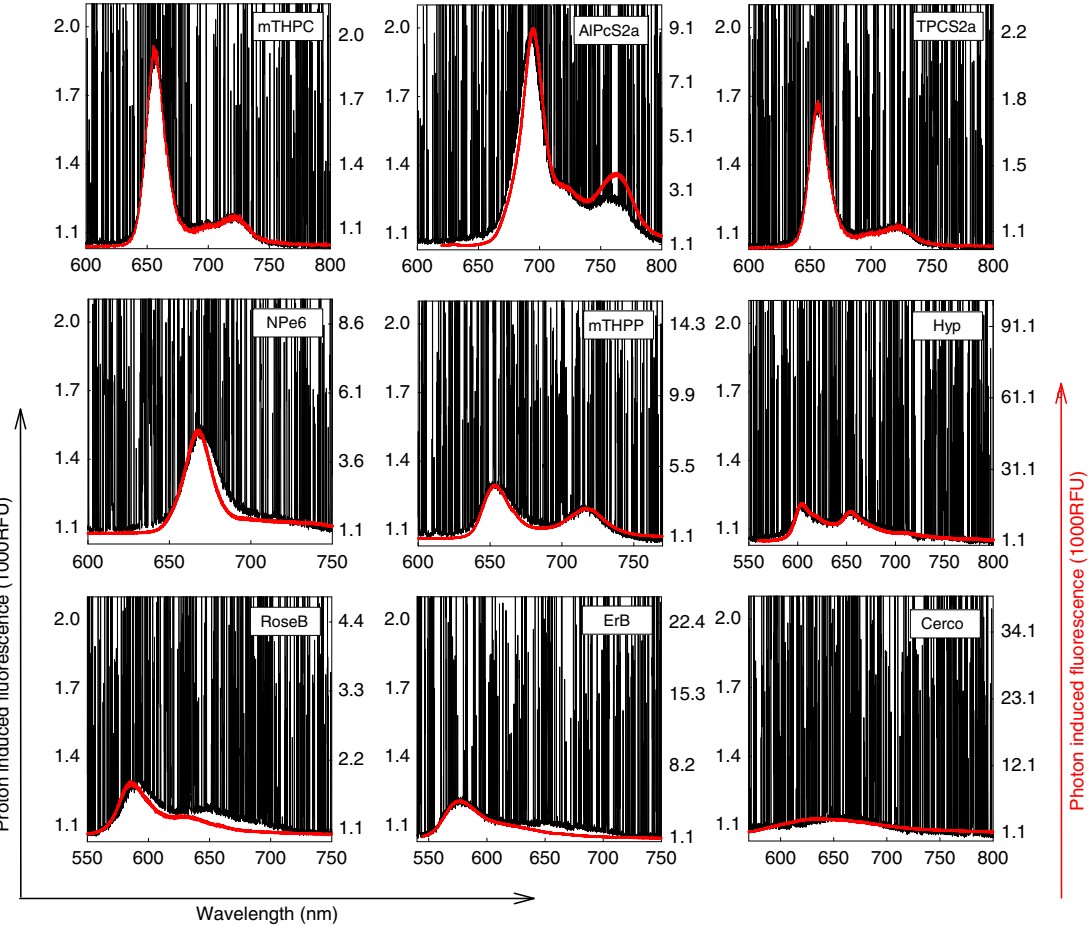

**Fig. 3** Spectral profiles of proton-induced fluorescence. Emission spectra for nine photosensitisers (PSs) obtained under proton excitation are shown in separate panels in black. All PSs were studied at 10 μM concentration in dimethyl sulfoxide (DMSO). The corresponding spectra of the PSs under light excitation were recorded from 10 μM DMSO solutions and are shown in red

oxygen by PpIX was also shown through the registration of its singlet-oxygen-derived photoproduct. We followed the PpIX photoproduct evolution by using fluorescence spectroscopy after proton irradiation, in the absence and presence of 1,4-diazabicyclo [2.2.2]octane (DABCO), a specific singlet oxygen scavenger[9]. In contrast to the absence of DABCO, in the presence of DABCO no photoproduct could be detected within our irradiation regime (Fig. 6c). In addition, we studied the photo- vs. proton-degradation of PpIX fluorescence (Fig. 6d). In the case of light irradiation (Xenon lamp 350–750 nm, 16 mW × cm$^{-1}$), the fluorescence was only moderately reduced through photobleaching (max ~20%, Fig. 6d), and this was almost completely abrogated by the application of DABCO. However, in proton irradiation, the fluorescence loss was profound (max ~ 80%, Fig. 6d), and DABCO only partially protected (max ~25%, Fig. 6d, Supplementary Fig. 3) against this fluorescence depletion.

**Proton-dynamic therapy in 2D cell cultures**. Having established the proof of principle of the proton-dynamic technology in PS solutions and gels, we tested its efficacy on 2D cell cultures. Three human GBM cell lines were employed for these pilot experiments (M059K, T98G and U87), and their differential death following proton irradiation was assayed in the absence and presence of the potent PS Cerco[10]. The positive results for M059K and T98G cells are shown in Fig. 7; no differential proton-dynamic cell death was observed in the two U87 experiments performed.

In the case of M059K cells, as seen in the representative results (Fig. 7a), the differential death from the addition of the PS to the cells before irradiation was statistically significant for 2 ($P =$ 0.045) and 5 Gy ($P = 0.02$), while for 10 Gy we had marginal statistical significance ($P = 0.065$). Correspondingly, in the case of T98G we had statistically significant enhancement of cytotoxicity due to proton-dynamic processes for 10 Gy ($P = 0.04$) and 20 Gy ($P = 0.006$). In the case of 30 Gy for T98G and 15 Gy for M059K, we could not perform any statistical analyses as these doses were tested once only due to their high toxicity. In Fig. 7c we have represented the differential gain in cytotoxicity due to proton-dynamic effects, i.e. due to the addition of PS prior to irradiation. From these data, we can see that the overall mean of gain from all five M059K experiments and for all doses is ~19%, while the median is ~16%. The corresponding values for T98G (4 experiments) were ~17% (mean) and ~10% (median). These results suggest that the M059K cells were more sensitive to the proton-dynamic effect at lower proton doses (2–10 Gy), while T98G cells were more responsive at higher radiation doses (10–30 Gy). Finally, as stated earlier, U87 cells were not responsive to proton-dynamic death at any of the doses of irradiation administered (data not shown).

## Discussion
To the best of our knowledge, the excitation of PSs used in PDT by accelerated protons is a novel concept. This study was inspired

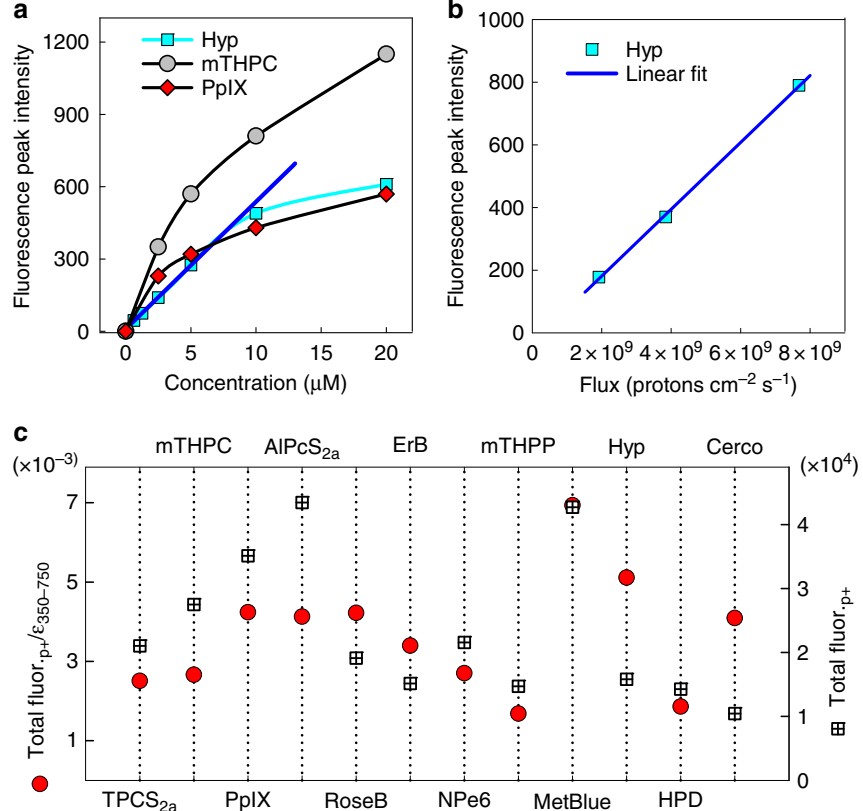

**Fig. 4** Proton-induced fluorescence dependence. Proton-induced fluorescence dependence on photosensitiser (PS) concentration, proton flux and proton-induced fluorescence efficiency by PS. **a** Concentration dependence of the peak fluorescence intensity of Hyp, PpIX and mTHPC. **b** Fluorescence intensity dependence on proton fluence for Hyp (10 μM, dimethyl sulfoxide). **c** Comparison of the calculated proton-induced fluorescence efficiency between selected PSs scaled to methylene blue. The black squares show the integrated proton-induced fluorescence for each PS; the red circles show the same values normalised by the integrated extinction coefficient (350–750 nm) for each PS. Source data for c are provided as a Source Data file

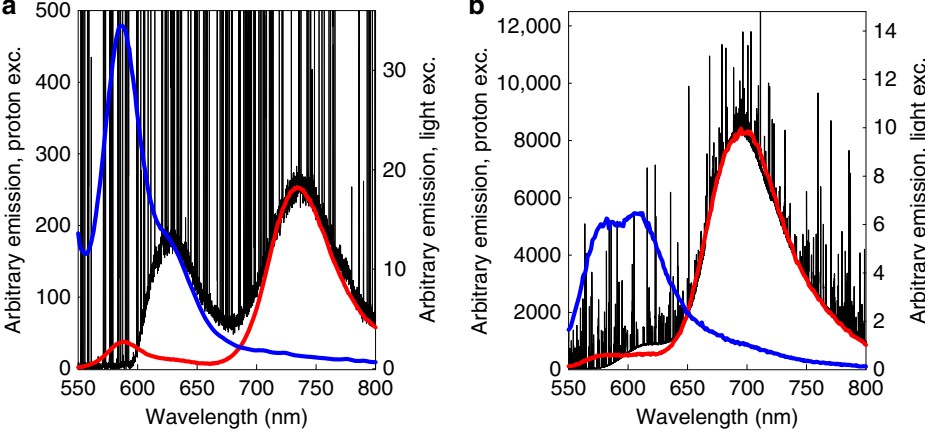

**Fig. 5** Proton-induced emission in dry photosensitiser (PS) gels. **a** Rose Bengal (RoseB) gel and **b** Erythrosin B (ErB) gel. The black lines correspond to the proton-induced emission of the gels (1 g gelatin (per 4 mL water, 500 μM PS), the blue lines denote the fluorescence emission from photonic excitation and the red lines show photon-induced phosphorescence ($\lambda_{ex} = 530$ nm for RoseB and 470 nm for ErB)

by the work of Birks[11], who explored the scintillation of organic crystals, and in particular anthracene as a solid, under the excitation of various types of radiation, including protons. Similar scintillation studies for the detection, discrimination and dosimetry of proton beams have used proton excitation but were mainly on rare-earth complexes in crystal form[12–14] and plastic-based scintillators[15].

We looked for the first time at the excitation and activation of a panel of PSs in solutions and gels by accelerated protons through the registration of their fluorescence and singlet oxygen

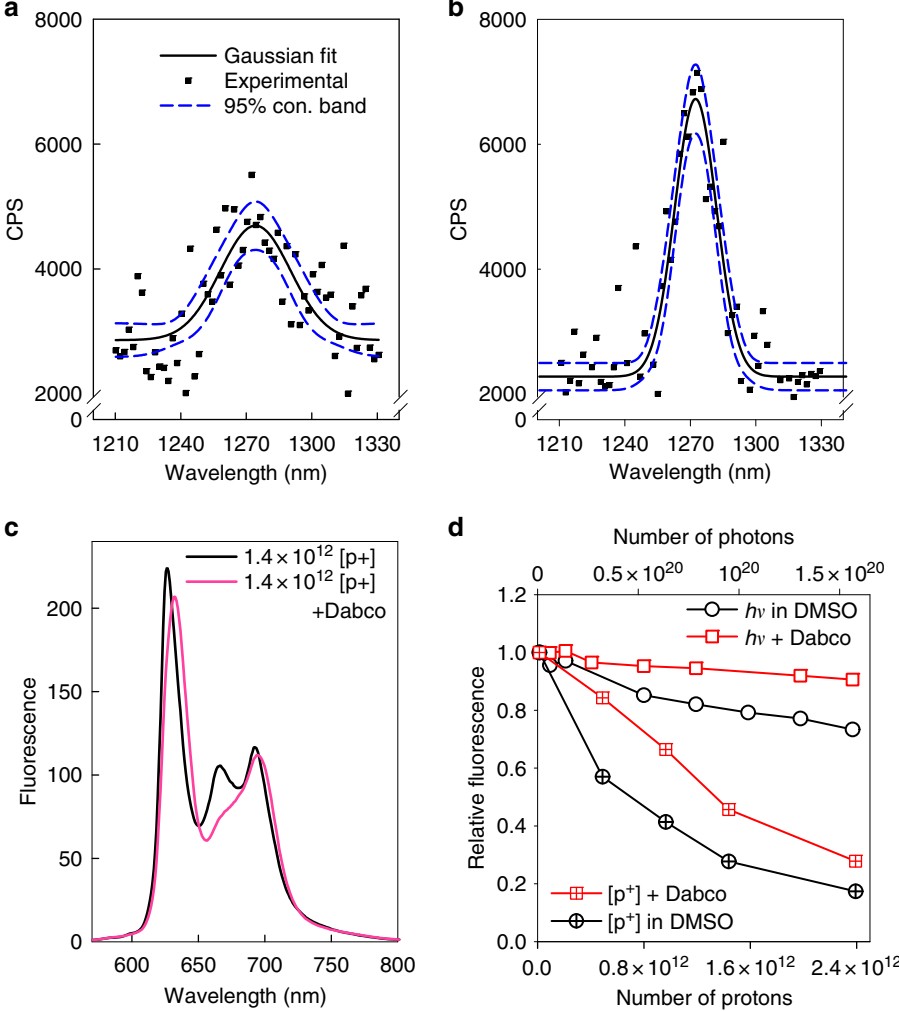

**Fig. 6** Proton-beam induced singlet oxygen and fluorescence degradation. Singlet oxygen phosphorescence, registered during proton beam irradiation at a fluence rate of $3.2 \times 10^9$ protons cm$^{-2}$ sec$^{-1}$ in acetone d6 solutions of **a** Erythrosin B (ErB) (60 μM) and **b** Rose Bengal (RoseB) (50 μM). **c** Protoporphyrin IX (PpIX) photoproduct (photoprotoporphyrin) production in the presence and absence of the singlet oxygen-specific scavenger 1,4-diazabicyclo[2.2.2] octane (DABCO) (200 mM) following proton irradiation of 50 μM PpIX in dimethyl sulfoxide (DMSO). The spectra were taken with a spectrofluorometer at 405 nm excitation. **d** Time evolution of PpIX (50 μM, DMSO) fluorescence intensity at 635 nm against irradiation dose either by a proton beam (crossed symbols) or by light (Xenon lamp, 350–750 nm, open symbols), in the presence (red squares) or absence (black circles) of DABCO (200 mM). In both cases, data were normalised to the unirradiated 635 nm maximum. The fluorescence data of panels **c** and **d** were acquired at 405 nm excitation

generation. The purpose of the study was to establish the basis for a new treatment concomitant with proton therapy, using PSs to destroy cancerous lesions by the generation of singlet oxygen in synergy to ionisations caused by the protons.

There have been a few attempts to use radiosensitisation by other forms of radiation such as γ-radiation and X-rays in combination with PSs. In those studies, PSs were employed as adjuvants with and without light excitation, while there was no indication of direct radiodynamic sensitisation of the PSs, e.g. through the registration of concomitant singlet oxygen generation[16,17].

Protons have a clear advantage over other radiation types such as γ-radiation and X-rays (photons) or β-radiation (electrons). As they traverse matter, they lose energy through interactions with the electrons they encounter on their course. Initially, the energy loss is almost constant with depth, up to the point where the cross section of interaction with electrons becomes very high and a high amount of energy is released in a spatially confined Bragg peak. This is in contrast with β- and γ- radiation and X-rays,

where the highest energy loss is upon entrance and the energy loss subsequently decreases with distance. This significant discrepancy allows the high-energy dissipation of protons to be tuned to a specific spatial position, such as the cancer lesion site in proton therapy. In addition, there is no energy left in the proton beam after the Bragg peak that could cause damage to normal tissue.

Our main hypotheses were that the use of a PS would: (i) accelerate the occurrence of the Bragg peak in places of high PS concentration (i.e. cancer lesions) and (ii) generate fluorescence and singlet oxygen upon proton excitation, which is the main PDT cytotoxic mediator. With respect to the first hypothesis, the analysis of the images in Fig. 2 clearly shows that the profile of proton-induced fluorescence is similar to the proton dose deposition profile (Supplementary Fig. 1). However, the spatial fluorescence pattern is distorted (Supplementary Fig. 1, solid lines), showing the induction of high-intensity fluorescence earlier than the Bragg peak-like maximum (Supplementary Fig. 1, black dots—simulated). This suggests that the PS may indeed be

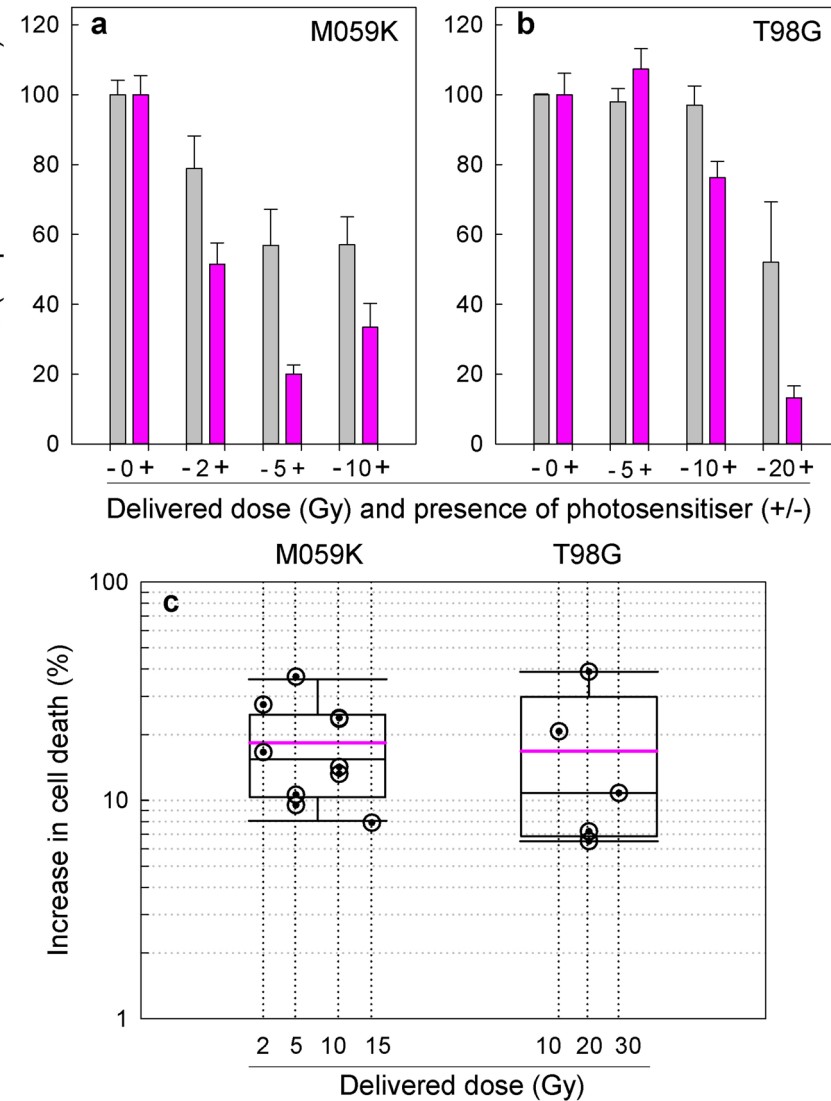

**Fig. 7** Proof of principle of proton-dynamic cell death in glioblastoma cell lines using cercosporin as the photosensitiser (PS). Differential cell death of M059K (**a**) and T98G (**b**) human glioblastoma cell lines, for varying radiation doses (2–5 Gy) with/without PS (cercosporin 4–10 μM, 4 h incubation), vs. their respective controls. The grey bars depict cell groups without PS while the magenta bars depict cell groups with PS. These results showcase the best experiments so far (3 parallels). Errors are represented as standard errors of the mean. **c** Box plot illustrating the increase in proton-dynamic differential cell death (cells without PS vs with PS) from all experiments conducted on M059K and T98G cells. The magenta line shows the mean of all experiments while the black line represents the median. The error bars represent s.e.m. Source data for **a**, **b** are provided as Source Data files

accelerating the high-energy depositions prior to the Bragg peak by highly increasing the interaction cross sections (ErB, 110 μM, Supplementary Fig. 1, red trace). The concentration of 110 μM in solutions may, at first sight, seem excessive. If we, however, consider that the average volume of normal or neoplastic mononuclear cells is ~200 μm$^3$ or about 0.2 femtolitres ($2 \times 10^{-16}$ L)[18], it would take the internalisation of a mere 13000 ErB molecules to reach 110 μM intracellularly. Furthermore, if the PS had an affinity for subcellular organelles, e.g. the mitochondria, then the same concentration might be reached by 10–100 molecules per mitochondrion.

Regarding the second hypothesis, we have shown that accelerated protons were able to excite a panel of PSs to produce fluorescence and generate singlet oxygen. As protons traverse the medium with the PS, they undergo Coulomb interactions with delocalised (π) electrons, either ionising these electrons or exciting them to their higher, metastable electronic states from which they return to the ground state emitting fluorescence. While in the singlet excited state, however, they can undergo intersystem crossing to the triplet state where they can interact with molecular oxygen (originally in triplet state). Through this collisional interaction both the PS and the oxygen undergo spin flip to their respective singlet states, and thus cytotoxic singlet oxygen is formed (Supplementary Fig. 4).

Two main factors determine the probability for ionisation or excitation of π electrons upon interaction with protons: (i) the kinetic energy of protons and ii) the distance between the proton and the delocalised electron. With respect to kinetic energy, the faster the protons (higher initial energy), the shorter the time of interaction with the electrons and consequently the lower the probabilities, predominantly for electron ionisation, but also for excitation. The distance between the passing proton and the electron also determines the intensity of the Coulomb interaction, as this is (in the classical frame) inversely proportional to the

square of the distance. The closer the proton passes by the electron the stronger the probability for ionisation, whereas as the distance increases the probability of excitation vs. ionisation increases. As the Bragg peak is reached, the kinetic energy of the protons decreases to a point where the interaction time with bystander π electrons increases to an optimal value, giving a high cross section for interactions, leading both to electronic ionisations and excitations. This is probably the reason for the broadened profile of the fluorescence (Supplementary Fig. 1 red trace) with respect to the simulated Bragg peak in the absence of PS (Supplementary Fig. 1 black dotted trace). The addition of the PS increases the density of π electrons quite profoundly compared to the plain medium (solvent), and thus the probability for excitation is much higher even before the onset of the Bragg peak. This opens up the possibility of lesion-specific cancer therapy, due to the specific accumulation of PSs in tumours, especially in cases of brain cancers (e.g. GBM) in which the blood-brain barrier has been breached, making cancer-specific accumulation even higher. This has been previously shown for the PS PpIX, which is endogenously generated following the administration of 5-aminolevulinic acid (5-ALA)[19]. In fact, 5-ALA derived PpIX fluorescence has been found to have higher diagnostic accuracy for GBM than gadolinium-enhanced magnetic resonance imaging contrast[20], and PpIX has been used intraoperatively to define the of GBM lesions resection margins[21]. Even though PpIX has the advantage of being endogenously synthesised in the haem biosynthetic cycle, a remarkable selectivity for GBM has also been shown in the clinical context for other external PSs such as temoporfin (mTHPC)[22]. In light of this information, the specific accumulation of PSs in the cancerous lesion can expedite the onset of a Bragg peak on the lesion site and use the ROS derived from the proton-induced PS activation to kill the cancer cells in synergy with proton therapy.

In our study, we registered efficient fluorescence production following the excitation of a panel of PSs by accelerated protons. The spectral profiles of proton-induced fluorescence in all cases matched closely the corresponding profiles obtained by photonic excitation at 532 nm using the same acquisition parameters and registration system (CCD-spectrograph). The spikes seen in every luminescence registration in the present work are artefacts from the ionising radiation, as it is well known that CCDs are very sensitive even to cosmic rays. These spikes represent visible light scintillations in response to ionising radiation, which can be present both in charge-coupled silicon devices and metal-oxide semiconductors.

Looking at the luminescence from dry gels (Fig. 5), it is obvious that, in the absence of water and hence oxygenation but also collisional quenching, the proton-induced phosphorescence emission (verified spectrally by photonic excitation) is quite prominent, since the triplet state lifetime is significantly extended by the ErB and RoseB examples. In turn, this indicates that the triplet state is populated also in solutions, yet is very quickly quenched by collisions with ambient oxygen, which may lead to singlet oxygen formation via energy transfer and mutual spin flip. This is evidence that the PSs occupy their triplet state when excited by accelerated protons and is also an initial indication that from there they are available to interact with molecular oxygen to generate singlet oxygen. The fact that the proton-induced phosphorescence of ErB was 40-fold that of RoseB, albeit their light-induced phosphorescences and proton-induced fluorescences were comparable, might suggest that the spin-forbidden direct pumping to the triplet state in the case of ErB is much more favourable.

The registration of singlet oxygen from proton excitation of PSs is a very encouraging sign for the feasibility of a hybrid synergistic treatment using proton therapy and ROS. The level of singlet oxygen production is quite low, but we have in the past registered cytotoxic results from chemiluminescence photo-sensitisation[23], in which singlet oxygen production was significantly lower. Additionally, in our experimental setup the proton energies were quite small (~15–16 MeV), and could only achieve a small penetration into matter (~2 mm in solvent). However, in therapeutic facilities, where energies in excess of 200 MeV are used, tissue penetration is quite substantial, reaching much deeper than light (Supplementary Fig. 5).

The singlet oxygen-associated PpIX photoproduct is another way of registering singlet oxygen generation[24]. Using the data initially presented in Fig. 6d, we further analysed the formation of the PpIX photoproduct by light and proton irradiation. Naturally, the photoproduct undergoes bleaching much like the parent molecule (PpIX), since it is also a quite potent PS[25]. By making the assumption that the photoproduct degradation is similar to that of PpIX, we have deconvoluted the photoproduct's evolution from its destruction in both proton and photon irradiation (Supplementary Fig. 6). In protonic and photonic excitation and for the fluences studied, the photoproduct formation is comparable between photon and proton irradiation and higher in the case of protons. Since the photoproduct formation is directly related to the amount of singlet oxygen generated[26], it can be used as dosimetry in PpIX PDT.

The fluorescence depletion versus irradiation time was much more efficient in the case of protons, as ~$10^{11}$ protons conferred the same depletion in fluorescence as ~$10^{20}$ photons. Moreover, in photonic irradiation, DABCO conferred an almost complete abrogation of the fluorescence bleaching, whilst in proton irradiation only a partial recovery of the fluorescence degradation was observed, despite the fact that DABCO completely suppressed the photoproduct formation (Fig. 6c). This implies that, although singlet oxygen is also partly involved in the fluorescence degradation by proton bombardment, mechanisms other than PS bleaching (mainly electron ionisation) are the predominant mechanisms of fluorescence loss and chemical modification.

In the clinic, prescription doses are based on the photon experience, adjusted for the radiobiological effectiveness of the protons (relative biological effectiveness = 1.1). In this sense, if we consider a clinically relevant daily dose of 50–60 Gy in 25–30 fractions, in our settings this would be equivalent to a continuous irradiation time of ~90 s (~0.66 Gy sec$^{-1}$ within the Bragg peak). This corresponds to a mere ~10% fluorescence loss (Fig. 6d). Moreover, in the cells our therapeutic doses varied from 2–30 Gy, which is much lower than in the clinical setting. Therefore, we do not expect the proton destruction of the PSs to be a limiting factor in the clinic.

In addition to solutions and gels, in the present study, we also present proof-of-principle in 2D cell cultures. From the results presented herein, we can see that the proton-dynamic effect is different in the three GBM cell lines investigated. The best overall effect was observed in the M059K cells for lower radiation doses (2–10 Gy), while in T98G the best effect was observed at higher doses (≥10 Gy). This is most probably due to the fact that the M059K cells are considerably more sensitive to proton irradiation (without PS) than T98G, and hence at high radiation-induced cytotoxicities in M059K it is difficult to differentiate the proton-dynamic effect from radiation cytotoxicity. U87, on the other hand, did not produce any detectable proton-dynamic cytotoxicity. This, however, can be related to our previous results[10] showing that the cercosporin uptake of U87 cells was profoundly lower (~3 times) than that of T98G. We have reproduced these results to also include M059K cells (Supplementary Fig. 7). These data confirm the ~3-times higher uptake of cerco by T98G in relation to U87, while M059K exhibit a ~2-fold higher uptake than U87 cells.

The results presented in Fig. 7 show a ~20% additive proton-dynamic effect in the best case, depending on the tumour cell type. Although this enhancement is already very significant and could benefit patients with more effective elimination of the cancer cells in deep-lying lesions such as GBM, the full potential could be substantially higher than the results presented in 2D cultures. Cell monolayers have very small dimensions in the plane of proton beam incidence and hence require very precise positioning (in the range of μM) for optimisation of the proton-dynamic cytotoxicity. In our future research, we plan to evaluate the proton-dynamic efficacy in 3D cell cultures, as we expect that the depth effect will be seen more clearly and hence the optimal efficacy depth will be determined without any need for positioning in relation to the Bragg peak profile.

The determination of the Bragg peak position for cell experiments had to be done independently for each experiment in our setting. In addition, as expected for research-grade cyclotrons, we encountered variant beam conditions between experiments, which necessitated readjustment of the experimental parameters. These variations can account for the discrepancies in the outcomes of separate experiments, which in turn affect the overall statistical profiles of the average outcomes. For this reason, we plan to translate our research to clinical setups in big proton centres, where the experimental conditions are accurately controlled.

In this seminal study, we present the generation of fluorescence and associated singlet oxygen from the excitation of PSs in solutions and gels by accelerated protons. We also provide proof of principle of the efficacy of the proton-dynamic effects on GBM cell lines. Our results may lay the foundations for a new hybrid therapy, adjuvant to proton therapy, as the ROS generated by this interaction could destroy cancer cells in synergy with ionising radiation. Such a treatment would require proton beam irradiation at existing proton therapy facilities, with the additional administration of a PS, and would benefit from the therapeutic effects of proton therapy and PDT, without the requirement of an external light source. We are currently at the very beginning of this journey; however, we envisage the therapy could be integrated into current proton therapy practices fairly quickly (10–15 years). The next steps require the treatment to be validated in 3D cultures and tests on animals. At the same time an optimal, customised PS has to be identified, synthesised, validated and approved for clinical use.

Our findings could have significant applications not only in medicine but also in other fields, such as radiation dosimetry.

## Methods

**Chemicals and reagents**. Dimethyl sulfoxide (DMSO), acetone d6, 1,4-diazabi-cyclo[2.2.2]octane (DABCO), thiazolyl blue tetrazolium bromide (MTT), coumarin 102, cercosporin (Cerco), hematoporphyrin (HPD), protoporphyrin IX (PpIX), temoporfin (mTHPC), hypericin (Hyp), erythrosin B (ErB), Rose Bengal (RoseB) and methylene blue (MetBlue) were purchased from Sigma-Aldrich Norway AS (Oslo, Norway). Amphinex® (TPCS2a) was kindly provided by PCI Biotech AS (Oslo, Norway). *Meso*-tetra(p-hydroxyphenyl)porphyrin (mTHPP), aluminium 1,4-di(sulfonyloxy)phthalocyanine (AlPcS2a) and *N*-aspartyl chlorin e6 (NPe6) were purchased from Frontier Scientific (Logan, UT, USA). 6,8-difluoro-7-hydroxy-4-methylcoumarin (DiFMU) was obtained from ThermoFisher Scientific (Oslo, Norway). The gelatin powder was regular food grade, of animal origin (swine), (Dr. Oetker Norge AS, Kolbotn, Norway).

**Proton irradiation**. Protons were accelerated by an MC-35 Scanditronix cyclotron (Scanditronix, Uppsala, Sweden). The proton energies were selected at 16 MeV by the analysing magnet. An ionization chamber (Advanced Markus type, PTW, Freiburg, Germany) was used for estimating the absorbed proton dose D in units of Gy as in a previously established beamline for radiobiological experiments[27]. The fluence Φ was calculated from the equation D = Φ × (S/ρ), where (S/ρ) is the mass stopping power. Proton energy and stopping power were estimated as a function of depth using the residual range method. The fluence rate entering the sample was $8 \times 10^8$ protons × cm$^{-2}$ × sec$^{-1}$ (5.3 Gy/s) in the case of spectral

measurements of fluorescence and phosphorescence as calculated. When capturing fluorescence images with a conventional Samsung NX1000 (Samsung Electronics Euro, Yateley, U.K.) camera, fluence rate was increased by a factor 40 to $32 \times 10^9$ protons × cm$^{-2}$ × sec$^{-1}$. The sample was placed in all cases at a distance of 30 cm from the beam exit window and the lateral beam profile at the sample was each time verified by the use of Gafchromic film EBT3 (Radiation Products Design Inc., Albertville, MN, USA). The experimental setup simplified to a linear layout comprising the cyclotron, the light-sealed sample compartment and light detection system is shown in Fig. 1.

**Proton-induced luminescence in solutions**. The sample was placed at the focal point of a Horiba iHR 320 grating spectrograph cavity (HORIBA Europe GmbH, Gothenburg, Sweden). The solutions measurements were performed in horizontally-oriented quartz or a vertically-placed plastic cuvette at a 90-degree angle. Protons entered the cuvette through a polystyrene film of 8 μm thickness. For fluorescence collection, a 1200 groove per mm diffraction grating blazed at 630 nm (visible—VIS) was used to disperse the light, which was consequently collected by a Synapse 1024 × 256 VOE CCD camera (Horiba Instruments Inc., Edison, NJ, USA). For the registration of singlet oxygen luminescence, a 600 groove/mm diffraction grating blazed at 1500 nm (near infrared—NIR) was used to disperse the light, which was collected by a Hamamatsu R5509-73 liquid nitrogen-cooled photomultiplier tube (PMT, Hamamatsu, Japan). With regard to the singlet oxygen luminescence, proton irradiation generated singlet oxygen in the aceton d6 solution without PS at a detectable level, which was however lower than the amount generated by the PS both in RB and ErB. In order to extract the pure singlet oxygen signal from the PS we had to (i) subtract the NIR PMT dark current and (ii) subtract the solvent contribution from that of the PS. The fluorescence images during proton irradiation were captured by a Samsung NX1000 camera (F-stop f/5.6, ISO speed 200) using a coloured long pass filter, with a cut-off wavelength of 610 nm placed in front of the objective. For the requirements of fluorescence imaging, the following PS concentrations were used: mTHPC 55 μM, PpIX 70 μM and ErB 110 μM. In the fluorescence spectral response measurements, all PSs were studied at 10 μM, while the UV dyes, coumarin 102 and 6,8-difluoro-7-hydroxy-4-methylcoumarin were studied at 200 μM and 5 mM, respectively. Singlet oxygen luminescence measurements were performed on 50 μM RB and 60 μM ErB, while PpIX photoproduct studies were conducted on a 50 μM solution.

**Dry gels**. 7.5 g of gelatin powder was dissolved into 30 mL of 80 °C Milli-Q® water by stirring. Aqueous stock solutions of either ErB or RoseB were mixed with the liquidised gelatin to reach 500 μM PS concentration before the water evaporated. The air-dried gels of 3 mm thickness were irradiated face-on with ~$10^9$ protons × cm$^{-2}$ × sec$^{-1}$, and the overall photonic emission was registered by the iHR 320/Synapse CCD. Following light excitation ($\lambda_{ex}$ = 530 nm for RoseB and 470 nm for ErB), the fluorescence and phosphorescence emission spectra were registered with the Varian Carry Eclipse spectrofluorometer (Agilent, Santa Clara, CA, USA), using a 5 ms gating time and 0.2 ms delay in the phosphorescence mode.

**Light irradiation**. Monochromatic excitation of PSs was facilitated by a 532 nm diode laser with variable output power (0–300 mW). The intensity of a collimated beam at the position of the sample was measured using an optic power meter, Gentec Solo2 (Gentec Electro-Optics Inc., Quebec, QC, Canada), equipped with a silicon photodetector, PH100 SiUV (Laser Components Nordic AS, Gothenburg, Sweden).

PpIX photoproduct formation under photonic excitation was achieved using a solar simulator, model 16S-300-002 (Solar Light Co., Inc., Glenside, PA, USA), equipped with a 150 W xenon arc lamp. KG-3 and appropriate dichroic filters were employed to ascertain a broadband spectrum of irradiation (350–750 nm). Spectral irradiance was measured using a portable Avantes AvaSpec 2048 × 14 FiberOptic spectrometer (Azpect Photonics AS, Sördertälje, Sweden). By integrating the spectral irradiance, a total irradiance of 16 mW × cm$^{-1}$ was obtained at the sample position. In both laser and Xenon lamp irradiation, the number of photons was calculated by multiplying the intensity of light at the particular wavelength by the exposure time in seconds and dividing by the energy of one single photon. In the case of the Xenon lamp, the number of photons at individual wavelengths was summed.

**Cell experiments**. $10^6$ cells (M059K CRL-2365™, T98G ATCC CRL-1690™, and U87 ATCC HTB-14™ human GBM cell lines, all purchased from and authenticated byATCC, Manassas, Virginia, USA), were seeded in 60 mm petri dishes 24 h prior to the experiments and incubated overnight at 37 °C in a 5% CO$_2$ humidified atmosphere. Before experiments, the cells were tested negative for mycoplasma using Mycoalert®, Mycoplasma detection kit (Lonza, Rockland, ME, USA). The cells were divided into groups according to the irradiation dose (0–30 Gy) and the presence or absence of PS (cercosporin 4–10 μM for 4 h). Following PS, incubation media were removed from the cells, and they were irradiated by protons through the petri dish underside, within the Bragg peak as determined by the combination of a transmission chamber (Monitor Chamber 7862, PTW, Freiburg, Germany) before the petri dish and an ionisation chamber (Advanced Markus® Electron Chamber, PTW, Freiburg, Germany) after the petri dish. The cells never remained

without media for longer than 10 min, while all controls were treated in the same manner, even when not irradiated. All cell groups were carefully shielded from ambient light, while the proton irradiations were performed in dark ambient surroundings. The dosimetry was performed with respect to the values obtained with the calibrated Markus® ionisation chamber[11]. During the irradiation, the cells were maintained at 37 °C using customised heated housing. Subsequently, fresh media were added to the cells and they were incubated for 20 h prior to the cytotoxicity assessment. The cytotoxicity was evaluated using a standard MTT assay: The cell media were replaced with 1 ml complete media containing 0.5 mg ml$^{-1}$ MTT. The cells were incubated for 1.5–2 h with the MTT media and then the media were removed and replaced with a 1 ml DMSO/dish. The dissolved formazan crystals were transferred to new plates that were read for absorbance, at 561 nm, by a Tecan spark M10 plate reader (Tecan Group Ltd., Männedorf, Switzerland). Blank values (wells with no cells) were in all cases subtracted. Cell viability was determined by comparison to the respective controls (irradiated cells without PS were compared to media controls, while PS-incubated and irradiated cells were compared to PS dark controls). In all experiments, each cell group consisted of at least 3 parallels.

**Flow cytometry**. U87, M059K and T98G cells were seeded in 60 mm-petri dishes ($10^6$ cells per dish) and incubated overnight in their normal media at 37 °C, in a 5% $CO_2$ humidified atmosphere. After 4-h incubation with cercosporin, cells were trypsinized, washed in PBS, resuspended in 500 μL PBS containing $Ca^{2+}$ and $Mg^{2+}$ and measured with flow excitation at 488 nm. The fluorescence was registered in the PerCP channel (685/35-nm bandpass filter) of an LSRII (BD) flow cytometer. Flow cytometry data were analysed in FlowJo v.7.6.1 (Treestar Inc., Ashland, OR)

**Software and data processing**. To depict the fluorescence intensity of selected PSs versus sample depth in 2D and 3D modes, two RGB (true colour) images were acquired (camera model Samsung NX1000, F-stop f/5.6, ISO speed 200, focal length 50 mm, resolution 5472 × 3648 and bit depth 24) of the PSs in the custom-made cuvette with and without proton irradiation. The two corresponding images were spatially superimposed on each other using MATLAB® (MathWorks, Natick, MA, USA). The illuminated true colour image was converted to grayscale by eliminating the hue and saturation information while retaining intensity information for each pixel and thereby showing intensity vs depth in the solution.

With reference to Supplementary Figs. 1 and 5, the geometrical scheme of the proton beam system was defined by using Flair v2.3-0 (FLUKA Advanced Interface). The absorbed dose profiles of 16 MeV (in DMSO) and 200 MeV protons (in a head phantom) were simulated in FLUKA v2011.2× -4. In both, the Gaussian energy spread was set to ±0.1% full width at half maximum (FWHM) with a Gaussian FWHM lateral spread of 10 mm. The profile of 800 nm light penetration into the skin was reproduced by using the AccuRT radiative transfer model (Geminor Inc., Maplewood, NJ, USA), in which total irradiance has both diffuse and direct light components.

**Computer code availability**. All computer codes used for data analysis are available from the corresponding author on reasonable request.

**Reporting summary**. Further information on research design is available in the Nature Research Reporting Summary linked to this article.

## Data availability
The datasets generated during and/or analysed during the current study are available from the corresponding author on reasonable request.

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

## Acknowledgements
The present work was financially supported by the South-Eastern Norway Regional Health Authority (Helse Sør Øst, project "protonic" no. 2017116). We would also like to thank Pawel A. Sobas, Victor Modamio and Jan C. Müller for the technical support with the cyclotron.

## Author contributions
M.G., Experimental design/experimental planning/ experimental work/ bespoke technical constructions for proton-dynamic experiments, data analysis, manuscript writing and figures production; M.arM., Experimental work/data analysis for proton-dynamic experiments, manuscript writing; K.B., experimental design/ data analysis for proton-dynamic experiments, manuscript writing; A.D., experimental work/image analysis for proton-dynamic experiments, Monte Carlo simulations, manuscript writing; M.atM., experimental work, manuscript editing; T.H.R., experimental work; E.B, bespoke technical constructions for proton-dynamic experiments; A.G., Experimental work (proton irradiation), manuscript editing; S.S., Experimental design input (cyclotron-irradiations),

manuscript editing, N.F.J.E., Experimental design input/experimental work (proton irradiations/ dosimetry), manuscript editing; E.M., Experimental design input/data analysis (proton irradiations/ dosimetry), manuscript editing; T.A.T., proton-dynamic concept, funding acquisition, project management, experimental design/experimental planning/ experimental work for proton-dynamic experiments, data analysis, manuscript workflow and writing, correspondence.

## Additional information

**Competing interests:** The authors declare no competing interests.

