## [Peer Review File · Nature Communications]

Reviewers' comments:

Reviewer #1 (Remarks to the Author):

The manuscript submitted by T.A.Theodossiou and co-workers is very important for the scientific community working on cancer imaging and therapy with photosensitizers. This beautiful work shows for the first time the potential of photosensitizers to be excited with accelerated protons, in order to produce singlet oxygen and luminescence, which can be an alternative to PDT or sonodynamic therapy. The manuscript is well written and well explained, and data support the conclusions. The work can therefore be published and highlighted in Nature Communications with minor modifications:

Figure 6d shows a dose-dependence degradation of PpIX upon excitation with accelerated protons. What is the effective dose of protons used in clinics for cancer therapy (it probably depends on the type of cancer) and what could be expected for the stability of the photosensitizer in vivo? A discussion could be added lines 346-355.

Experimental: line 161-173, although the solvent and concentrations of photosensitizers were mentioned in captions of figures 2 and 3, it would be interesting to write it again in the experimental part.

Typographical errors:

line 40 millimeters

line 44 loose

line 234 loose

Reviewer #2 (Remarks to the Author):

Photodynamic therapy has long demonstrated its clinical value but also the severe limitations imposed by the need of delivering light to deeply-seated lesions. The notion that one could use protons to overcome this fundamental barrier is an important contribution to the field and potentially a long-awaited breakthrough.

In their communication, the authors successfully demonstrate that this is feasible. Nevertheless, the report would benefit from a number of additions to increase its impact. Some comments follow:

- 1- All the photosensitizers used absorb in the visible part of the spectrum. Could one use protons to excite blue- or UV-absorbing photosensitizers that are typically more photo-stable than red or near-infrared ones?
- 2- Is it possible to excite oxygen directly with protons, without the need of a photosensitizer? Did the authors try to irradiate a photosensitizer-free sample?

3- Formation of phosphorescent triplet states is convincingly demonstrated. Did that occur by the usual intersystem crossing from originally excited singlet states or the triplet states can be populated directly by protons.

4- In lines 263ff it is assumed that the original excitation is to the singlet state. However, if the molecule is effectively ionized, electron-hole recombination ought to randomly produce one singlet and three triplets. Any hint that this may actually be happening?

5- The previous questions address the issue of how interaction with protons may lead to electronically-excited molecules. Can the authors comment on this and highlight the differences with respect to UV-Vis light excitation in terms of absorption cross sections, selection rules, etc.? That is, can they identify the desirable features of proton-sensitisers?

6- Proton-bleaching of photosensitisers seems to be more important than photobleaching (Figure 6d). The authors should expand their discussion on this point (line 337ff) on the implications for this for the potential therapeutic use of the proton-dynamic approach and potential strategies to circumvent it (see also question 1 above).

7- The authors comment extensively on the advantages of proton therapy as compared to X-Ray and gamma radiotherapies, particularly the ability of depositing the energy in a confined, tuneable region. It would be nice if they were able to demonstrate such spatial control of photosensitiser excitation, e.g., with their dry gel systems.

8- There are no results in cells or other biological systems. Were attempts made? Even if they failed, it is worth knowing they were attempted.

9- The authors' vision on a future roadmap for translating proton dynamic therapy to the clinical practice would be very welcome. Some degree of speculation is acceptable to rise enthusiasm.

Reviewer #3 (Remarks to the Author):

The paper describes the use of accelerated protons to generate fluorescence and reactive oxygen species. The authors state this as a first step towards a novel hybrid Cancer Treatment combining Proton radiotherapy with Singlet-Oxygen and other ROS mediated cytotoxicity.

In general, the manuscript describes the technical solution to provide such a potential treatment Option, but to claim this as a potentially effective treatment, one would expect at least proof-of-principal biological experiments on cells or so. Without this, the whole paper is hypothetical.

POINT BY POINT RESPONSES TO REVIEWERS COMMENTS NCOMMS-19-09680-T

The manuscript submitted by T.A.Theodossiou and co-workers is very important for the scientific community working on cancer imaging and therapy with photosensitizers. This beautiful work shows for the first time the potential of photosensitizers to be excited with accelerated protons, in order to produce singlet oxygen and luminescence, which can be an alternative to PDT or sonodynamic therapy. The manuscript is well written and well explained, and data support the conclusions. The work can therefore be published and highlighted in Nature Communications with minor modifications:

Figure 6d shows a dose-dependence degradation of PpIX upon excitation with accelerated protons. What is the effective dose of protons used in clinics for cancer therapy (it probably depends on the type of cancer) and what could be expected for the stability of the photosensitizer in vivo? A discussion could be added lines 346-355.

Response: We have now added a short discussion correlating our results to the clinical setting. This paragraph is highlighted in yellow (line 278-285)

Experimental: line 161-173, although the solvent and concentrations of photosensitizers were mentioned in captions of figures 2 and 3, it would be interesting to write it again in the experimental part.

Response: We have now included the solvent concentrations also in the materials and methods section lines: 377-382

Reviewer #2 (Remarks to the Author):

Photodynamic therapy has long demonstrated its clinical value but also the severe limitations imposed by the need of delivering light to deeply-seated lesions. The notion that one could use protons to overcome this fundamental barrier is an important contribution to the field and potentially a long-awaited breakthrough.

In their communication, the authors successfully demonstrate that this is feasible. Nevertheless, the report would benefit from a number of additions to increase its impact. Some comments follow:

1- All the photosensitizers used absorb in the visible part of the spectrum. Could one use protons to excite blue- or UV-absorbing photosensitizers that are typically more photo-stable than red or near-infrared ones?

*Response: We have now included two UV-dyes, namely coumarin 102 a laser dye with high fluorescence quantum yield and 6,8-Difluoro-7-Hydroxy-4-Methylcoumarin. The fluorescence spectral response is shown now in the new **supplementary figure 2** while the other supplementary figures have been renumbered (also in the manuscript text). We have added a paragraph in the results section (line 82-86) highlighted yellow, and also two new entries in the materials and methods (chemicals) also highlighted in yellow.*

2- Is it possible to excite oxygen directly with protons, without the need of a photosensitiser? Did the authors try to irradiate a photosensitiser-free sample?

*Response: Indeed the acetone-d₆ solvent produced singlet oxygen at a lower yet detectable level. In order to extract the pure singlet oxygen signal from the PS we had to i) subtract the NIR PMT dark current and ii) subtract the acetone-d₆ solvent contribution from that of the PS. In both cases we got a clear signal gain from the PS in singlet oxygen production (presented in Fig. 6A,B). We have now added a small paragraph in the materials and methods, **line 370-374**, highlighted in yellow.*

3- Formation of phosphorescent triplet states is convincingly demonstrated. Did that occur by the usual intersystem crossing from originally excited singlet states or the triplet states can be populated directly by protons.

Response: Looking at figures 3 and 5 (solutions vs. dry gels) we can see that the triplet states of either ErB or RB are not present in solutions (Fig. 3). This could, of course, be due to a fast quenching of the triplet states by oxygen which could also apply for light excitation. However we can see an efficient population of the singlet states in Fig. 3 (through fluorescence). Nevertheless we believe that since in the applied (clinical) settings there will always be ambient oxygen (also a requirement for a successful proton-dynamic outcome) then the triplet state will be mostly populated by the usual ISC pathway (since we can see that the singlet state is excited).

4- In lines 263ff it is assumed that the original excitation is to the singlet state. However, if the molecule is effectively ionized, electron-hole recombination ought to randomly produce one singlet and three triplets. Any hint that this may actually be happening?

Response: This is a very interesting question. We believe that this is also possibly happening, however it is very difficult to give some evidence about this. This question is partly related to the previous one, and if the triplet quenching is happening quite fast it is very difficult to document any further effects. Ionisation is a key process in the proton-dynamic however if there is a hint to some effect, this would be that the electron-hole recombination is not the main effect. As stated by the reviewer this process would be a 1(singlet) to 3(triplet) mechanism, so even in solutions if this was a dominant pathway then we would possibly see some phosphorescence, which, as stated in Q3, we don't.

5- The previous questions address the issue of how interaction with protons may lead to electronically-excited molecules. Can the authors comment on this and highlight the differences with respect to UV-Vis light excitation in terms of absorption cross sections, selection rules, etc.? That is, can they identify the desirable features of proton-sensitisers?

Response: We will attempt to answer this question only to the reviewers as at this moment we do not have the whole picture regarding the ideal proton sensitiser. Two main indications however are the stability in degradation (which would be limited due to the required aromaticity), and the large number of delocalised electrons for interaction (cross section). The classical rule for the photosensitisers for absorbance below 800 nm we guess still applies otherwise we only have direct

singlet oxygen generation, however the long wavelength rule does not apply as light is not involved any more.

6- Proton-bleaching of photosensitisers seems to be more important than photobleaching (Figure 6d). The authors should expand their discussion on this point (line 337ff) on the implications for this for the potential therapeutic use of the proton-dynamic approach and potential strategies to circumvent it (see also question 1 above).

Response: We have now added a short discussion correlating our results to the clinical setting. This paragraph is highlighted in yellow (line 278-285)

7- The authors comment extensively on the advantages of proton therapy as compared to X-Ray and gamma radiotherapies, particularly the ability of depositing the energy in a confined, tuneable region. It would be nice if they were able to demonstrate such spatial control of photosensitiser excitation, e.g., with their dry gel systems.

Response: We believe that the spatial control of the photosensitisers has been demonstrated through the fluorescence imaging by the camera in fig. 2. In this figure the fluorescence has been shown to follow a spatially controlled profile in depth, following the Bragg peak (pseudocolored contour map-fluorescence red-highest- on Bragg peak). With respect to the beam spread, as we mentioned in the text we have a research grade cyclotron with a beam divergence. In the clinical settings, the beam can either be scanned or pencil-like so the width considerations will be abolished.

8- There are no results in cells or other biological systems. Were attempts made? Even if they failed, it is worth knowing they were attempted.

Response: We have now included work on cells. This is presented in a new figure (Fig. 7) and also the appropriate texts have been introduced in The results section (line 126-146), Materials and methods section (line 407-431) and discussion (line .286-316). The abstract has been modified accordingly and so have the conclusions and last paragraph of introduction. All new additions and changes have been highlighted in yellow.

9- The authors' vision on a future roadmap for translating proton dynamic therapy to the clinical practice would be very welcome. Some degree of speculation is acceptable to rise enthusiasm.

We have now added a short paragraph in the discussion outlining our vision and roadmap to the clinic (line 325-329)

Reviewer #3 (Remarks to the Author):

The paper describes the use of accelerated protons to generate fluorescence and reactive oxygen species. The authors state this as a first step towards a novel hybrid Cancer Treatment combining Proton radiotherapy with Singlet-Oxygen and other ROS mediated cytotoxicity.

In general, the manuscript describes the technical solution to provide such a potential treatment Option, but to claim this as an potentially effective treatment, one would expect at least proof-of-principal biological experiments on cells or so. Without this, the whole paper is hypothetical.

*Response: Same as for reviewer 2 Q6: We have now included work on cells. This is presented in a new **figure (Fig. 7)** and also the appropriate texts have been introduced in The results section (**line 126-146**), Materials and methods section (**line 407-431**) and discussion(**line .286-316**). The abstract has been modified accordingly and so have the conclusions and last paragraph of introduction. All new additions and changes have been highlighted in yellow.*

Further additions:

*In addition to the reviewers comments we have added a new supplementary figure (**Supplementary Fig. 7**) to support our cell data discussion (**line 294-298**, highlighted yellow). This is in relation to the cell uptake of cercosporin (PS) and a new paragraph has also been added to the materials and methods (**lines 42-48**, highlighted yellow)*

We have also added an acknowledgements section and an authors' contributions section

REVIEWERS' COMMENTS:

Reviewer #1 (Remarks to the Author):

The manuscript submitted by T. Theodossiou and co-workers has been corrected following the advice of the referees and can be published as is. It also should be highlighted in the journal as the results are important for the scientific community.

Reviewer #2 (Remarks to the Author):

The authors have made a genuine effort to answer all the questions raised in my former review and carried out additional experimental work in response to those comments. The manuscript has been expanded along the lines suggested, hence this exchange of questions and answers will reach the final reader. The most important new result is the confirmation that cell death can be enhanced through the production of singlet oxygen generated by proton irradiation. Many scientific aspects remain to be discovered but the paper contains enough interesting results as to trigger the interest of the community to the very novel findings reported in this paper.

REVIEWERS' COMMENTS:

Reviewer #1 (Remarks to the Author):

The manuscript submitted by T. Theodossiou and co-workers has been corrected following the advice of the referees and can be published as is. It also should be highlighted in the journal as the results are important for the scientific community.

Reviewer #2 (Remarks to the Author):

The authors have made a genuine effort to answer all the questions raised in my former review and carried out additional experimental work in response to those comments. The manuscript has been expanded along the lines suggested, hence this exchange of questions and answers will reach the final reader. The most important new result is the confirmation that cell death can be enhanced through the production of singlet oxygen generated by proton irradiation. Many scientific aspects remain to be discovered but the paper contains enough interesting results as to trigger the interest of the community to the very novel findings reported in this paper.

RESPONSE TO THE REVIEWERS' COMMENTS:

We thank the reviewers for their challenging and interesting questions, criticism and recommendations.